# Using Cholesterol-Loaded Cyclodextrin to Improve Cryo-Survivability and Reduce Capacitation-Like Changes in Gender-Ablated Jersey Semen

**DOI:** 10.3390/ani15142038

**Published:** 2025-07-11

**Authors:** Ahmed S. Aly, Kevin J. Rozeboom, John J. Parrish

**Affiliations:** 1Department of Animal and Dairy Sciences, University of Wisconsin-Madison, Madison, WI 53705, USA; a_yousef129@agr.asu.edu.eg; 2Department of Animal Production, Faculty of Agriculture, Ain Shams University, Cairo 11241, Egypt; 3Genus PLC & ABS Global, De Forest, WI 53532, USA; kevin.rozeboom@genusplc.com

**Keywords:** cholesterol-loaded cyclodextrin, gender ablation technology, Jersey, cryopreservation, capacitation-like changes, flow cytometry, oviduct culture, in vitro fertilization

## Abstract

Sex-selected semen is now an integral feature of dairy cattle production and is essential to profitability and sustaining productivity. The sexing procedures and subsequent cryopreservation can damage sperm membranes, lower sperm lifespan, and ultimately reduce the fertility of sex-selected semen when compared to conventional semen. This study aimed to improve the quality of gender-ablated semen. Our approach was to increase the cholesterol content of sperm plasma membranes to reduce membrane damage and keep them intact until insemination. Adding cholesterol to sperm cells did not increase sperm motility. However, the portion of sperm cells with intact acrosomes and membranes significantly increased from 28.9 ± 1.2% to 34.1 ± 1.2% after adding cholesterol. Mitochondrial activity, maturation progress, and binding ability to oviduct cells (required for sperm lifespan inside the cow reproductive system) were maintained. Cholesterol treatment did not delay fertilization time but significantly increased fertilizing ability. After 12 h of co-incubation with oocytes, the fertilization rate rose to 82 ± 3% in the cholesterol-treated group, compared to 74 ± 3% in the control group (*p* < 0.05). Overall, the addition of cholesterol added minimal improvement to the quality of post-thaw gender-ablated semen, and further studies are still needed to maximize the benefits of cholesterol.

## 1. Introduction

The sex-selected semen can greatly benefit both dairy and beef breeders by enabling the production of calves of the desired sex, which can accelerate genetic gain and maximize profitability [1,2]. Currently, there are two established methods for producing sex-selected semen: gender ablation technology and the separation method [3,4,5]. Both technologies rely on staining sperm cells with Hoechst 33342 to detect differences in DNA content. The X-chromosome-bearing sperm typically has approximately 3.8–4% more DNA than Y-chromosome-bearing sperm, which enables differentiation between the two sexes [5,6,7]. After identifying the amount of DNA, sperm cells are separated using either gender ablation technology, where the undesired sex is destroyed using laser beams [8], or based on electric charge in the flow cytometry-based method [7]. The present study will utilize gender-ablated semen produced through gender ablation technology (Sexcel, ABS Global [ABS], Deforest, WI, USA), which was recently introduced in 2016 [4,8]. Gender ablation technology avoids sperm droplets, electric current, and high pressure, ensuring continuous stream microfluidics, which could be gentler to sperm cells, as described by [9]. Sexcel exhibits an average sperm sex ratio (purity) of 87.2% (desired/undesired sex) and a sex ratio of fetuses ranging from 84.6% to 92% [4,9,10].

Although there are benefits of sex-selected semen for both farmers and industry, it has many limitations. Sperm cells are exposed to multiple steps during sex selection that contribute to more handling from staining and incubation, orientation, high dilution rate, pressure, temperature, different media, exposure to laser, centrifugation, and ultimately longer holding time before freezing when compared to conventional frozen semen [4,6,8,11]. These processes likely cause more damage to the sperm plasma membrane and DNA and can induce capacitation-like changes [12,13,14]. Therefore, the conception rate (CR) of gender-ablated semen is lower when compared with conventional semen [4,8]. In three fertility trials using U.S. virgin dairy heifers gender-ablated semen was compared with non-sex-selected semen. The best results from gender-ablated semen were obtained in the third trial, with a fertility rate of 64.1 ± 1.6% for conventional semen vs. 48.3 ± 1.7% for Sexcel [4]. This means that under the best conditions, Sexcel fertility was 75% of that obtained by conventional semen. Similar results (67% for conventional vs. 52% for Sexcel) were obtained by [15]. Sex-selected sperm cells are fragile and should be handled carefully to achieve satisfactory CR. Thus, when sex-selected semen is used, producers should utilize high management and breeding protocols and select specific body condition scores and highly fertile groups, mainly heifers [16,17]. Sex-selected semen conception rates vary significantly between herds and individual bulls [17,18]. Maicas et al. [18] also concluded that incorporating checks for capacitation-like changes in post-sexing quality control could enhance fertility outcomes. This practice would enable the rejection of affected ejaculates or bulls before distributing semen straws.

Cholesterol is a hydrophobic molecule and is insoluble in water. Thus, cyclodextrin can be utilized to solubilize cholesterol by forming water-soluble inclusion complexes. The incorporation of methyl or hydroxypropyl groups enhances cyclodextrin’s ability to dissolve hydrophobic molecules [19,20]. The addition of cholesterol stabilizes the sperm plasma membrane, enhances thermo-resistance, compensates for cholesterol efflux during cryopreservation, and improves post-thaw sperm motility and viability [20,21]. The impact of cholesterol on fertility and capacitation status is controversial [20,22], with many scientists reporting lower capacitation-like change levels for cholesterol-treated sperm. Capacitation is a broad and complex process that includes numerous biochemical maturational changes at the levels of the sperm head, middle piece, and tail, which are not yet fully understood [23,24]. Consequently, evaluating capacitation varies significantly from lab to lab due to the numerous approaches for capacitation induction [22] and detection methods [23,25,26,27,28].

Flow cytometry analyses utilize large numbers of sperm cells, making it more accurate, sensitive, and reliable in evaluating capacitation-like changes when compared to fluorescence microscopy approaches [29,30]. Flow cytometry analyses in the present study involve simultaneous assessment of sperm viability using propidium iodide (PI) and acrosome integrity using FITC-PNA [31,32]. Sperm cells with intact membranes are not permeable to PI and thus do not stain. Using a combination of Merocyanine 540 (M540), Yo-Pro-1 (YP), along with Hoechst (to gate out non-sperm events), was found to be a robust approach in evaluating membrane stability and capacitation-like changes in bull sperm [33]. The YP infiltrates cells with increased permeability of pannexin-gated channels, which occurs before complete membrane integrity loss, allowing earlier detection of cell death compared to PI [34,35]. During capacitation, sperm cells lose glycoproteins, which increases membrane fluidity and destabilization [36], leading to more M540 intercalating by the sperm membrane [35,37]. Thus, YP and M540 can track membrane changes due to cryoinjuries, with YP indicating loss of membrane integrity and selective permeability, and M540 identifying early capacitation changes in lipid packaging and distribution within the sperm membrane. This also helps determine if cholesterol modification of sperm plasma membranes has occurred.

Another approach for tracking capacitation progress is monitoring intracellular calcium. Capacitated cells have a higher ability to uptake calcium from the surrounding media, accumulate it into their acrosome and cytoplasm [24,38], and undergo acrosome reaction if induced [22,39]. Mitochondrial potential, as evaluated using JC-1, can reflect sperm cell activity. Active mitochondria aggregate JC-1 and emit orange fluorescence, while less functional ones accumulate lower JC-1 concentrations, which fluoresce green. The ratio between these colors indicates the activity level of sperm mitochondria [30].

Binding sperm cells to oviduct epithelial cells in the isthmus region is vital for maintaining sperm viability and preventing premature capacitation [40,41]. Cryopreservation may lead to surface alterations that may impact the ability of sperm cells to bind to oviduct cells [42]. Un-capacitated sperm cells display better ability to interact with oviduct cells [43]. Therefore, binding to oviduct cells can indicate sperm capacitation status, with reduced binding as capacitation progresses. Binding to oviductal cells is essential for timely capacitation in the cow reproductive tract, which ensures a better chance of sperm cells fertilizing an oocyte. Capacitation is considered complete when sperm penetrates a zona-intact oocyte, fuses with the oolemma, and decondenses in the oocyte cytoplasm [24,44]. Thus, the capacitation window can be tracked based on the penetration time when sperm cells are induced to capacitate in vitro [44,45]. This is essential to determine any delays in capacitation due to the CLC treatment.

The goal of this study was to improve post-thaw survivability and reduce the capacitation-like changes in gender-ablated semen. This is expected to enhance fertility outcomes and reduce fertility variability between herds and bulls. Before freezing, semen straws are filled with 2.6 × 10^6^ motile sperm cells [4,8]; thus, it is critical to improve sperm cryo-survivability to increase the number of post-thaw viable motile sperm cells. Enhanced semen quality would also reduce the need for specific usage recommendations for farmers and, in turn, more flexibility when gender-ablated semen is used. The target is to increase the cholesterol content of sperm plasma membranes. The approach modifies the sperm plasma membrane [22], the most affected part during sex selection and cryopreservation [12,46,47]. This marks the first time attempting CLC with gender-ablated semen. The hypotheses to be tested include that CLC will increase post-thaw sperm motility, maintain the integrity of the sperm membrane, delay capacitation, enhance sperm binding ability to oviduct cells in culture, and increase fertilizing ability along with extending the capacitation window.

## 2. Materials and Methods

Bulls were housed in a commercial facility that collects semen from bulls at ABS Global (Animal Breeding Services, Genus PLC) in Rio, WI, USA. The facility follows the USDA guidelines for animal care, and bulls are under routine veterinary supervision. The Chemicals were procured from Sigma-Aldrich (St. Louis, MO, USA) or ThermoFisher (Pittsburgh, PA, USA) unless otherwise specified.

### 2.1. Cholesterol-Loaded Cyclodextrin Preparation

The CLC powder was prepared following the Purdy and Graham procedure [21,22,48]. Separate solutions of cholesterol and cyclodextrin were prepared by dissolving 0.2 g cholesterol in 1 mL of chloroform and 1 g methyl-β-cyclodextrin in 2 mL methanol, respectively, in two glass tubes. Subsequently, 0.450 mL of the cholesterol solution was combined with the 2 mL cyclodextrin solution, and the resulting mixture was vortexed until clear. Thus, the prepared cholesterol solution was sufficient for two tubes of the cyclodextrin solution. The mixture was then poured into a glass Petri dish and incubated at 37 °C for at least 24 h to remove solvents and yield CLC crystals [19]. The crystals were harvested and stored on the shelf at room temperature (around 22 °C). The CLC working solution was prepared by adding 0.50 g of CLC crystals to 1 mL of Bovine Gamete Media 3, abbreviated as BGM3 [24] at 37 °C, followed by vortexing for a minute. To enhance solubilization, the working solution was sonicated for 2 min. Two adjustments were made to the Purdy and Graham protocol: sonication and a tenfold increase in the concentration of the CLC working solution required to minimize the changes in the ABS red TALP (media used by ABS to dilute Hoechst-stained sperm cells before sex selection). In a preliminary study at Dr Parrish’s lab, the higher CLC concentration and sonication yielded similar motility results to the original Purdy and Graham protocol. Another reason for the sonication of the working solution is to prevent potential clumping in the gender ablation machine. Additionally, working with ABS entails storing the CLC working solution at 5 °C before adding it to the red TALP on the day of use.

### 2.2. Semen Collection, Experimental Design, and Data Adjustments

Semen samples were collected from four different Jersey bulls and underwent processing at ABS (Genus PLC) in Rio, WI, USA. Only sexable ejaculates according to the company’s standard operating procedures (SOP) were included. The semen was then diluted to a concentration of 200 × 10^6^ sperm cells per mL using a staining TALP media containing Hoechst (amount proprietary to ABS), followed by an incubation period of 60 min at 37 °C. Subsequently, the stained samples were divided into two aliquots. Aliquot 1 (control group) was processed according to the ABS SOP, where the stained cell reaction volume was mixed with red TALP containing 3% egg yolk at a ratio of 1:2, resulting in a final sperm concentration of 67 × 10^6^ sperm/mL. For aliquot 2, CLC was dissolved in BGM3 [24] and added to the red TALP. Direct addition of CLC into red TALP generated foam, and it was insoluble. Consequently, BGM3 was used to prepare the CLC working solution instead of red TALP. The mixture was used to dilute stained semen at a ratio of 1 stained cell volume to 2 red TALP volumes containing CLC. The CLC level in this group was 2 mg per 1 mL of diluted semen containing 67 × 10^6^ sperm per mL. The previous CLC level was determined as optimal based on a pilot study conducted using conventional semen. Following this, the samples were incubated at 37 °C for 15 min before gender ablation; all the subsequent steps were conducted according to ABS SOP. During gender ablation, sperm cells of the undesired sex were inactivated but remained in the sample [8]. Straws were filled with 2.6 million motile sperm of the desired sex before freezing. This established the baseline for adjusting viability, motility, and mitochondrial potential data. Therefore, post-thaw sperm concentration in million/mL for each bull and treatment group was separately calculated using CASA, then multiplied by straw volume to determine the total post-thaw motile cells per straw. This total was divided by 2.6 million to calculate the percentage of motile cells from the initial pack. Straws were coded to maintain blindness to both the observer and company personnel processing the samples. Frozen straws were shipped to UW-Madison in a liquid nitrogen vapor shipper tank. Upon arrival at the lab, they were stored in a liquid nitrogen tank at −196 °C until thawing.

### 2.3. Motion Analyses

The motion analyses of frozen-thawed sperm were determined using a computer-aided sperm analysis (HTM-IVOS CASA system, Hamilton-Thorn Research, Bedford, MA, USA), equipped with a warming stage at 37 °C. The settings of the CASA system were as described by Mocé and Graham, 2006 [48], which include analyzing 30 frames per second, minimum contrast = 50, minimum average path velocity = 25 µm/s, straightness = 80%, and nonmotile head size = 5. Semen straws were thawed at 37 °C for 30 s. A minimum of ten fields with at least 500 sperm per sample were evaluated at time points 0, 30, 60, and 120 min post-thaw. Total and progressive sperm motilities were assessed via CASA. In addition, visual motility was assessed at each time point using a phase-contrast microscope (200×) by observing five fields for each sample. Motility data were adjusted because sperm cells of the undesired sex are inactivated but remain in the straws. The baseline for the adjustment was 2.6 × 10^6^ motile sperm per straw, as explained above. Two technical replicates were conducted with the analysis of a different straw from the same freeze batch each time.

### 2.4. Flow Cytometric Analyses

The flow cytometry analyses were conducted at the UWCCC Flow Cytometry Laboratory, UW-Madison, using a BD LSRII flow cytometer (BD Biosciences, San Jose, CA, USA). Semen straws were thawed and diluted to 1 × 10^6^ sperm/mL with Tris diluent (Triladyl, Minitube USA, 13500/0250, Verona, WI, USA) except for calcium analyses, for which BGM3 [24] was used instead of Tris to provide a calcium source. Flow cytometric assays involved evaluating 10,000 Hoechst-positive events, gating out non-sperm events based on H33342 fluorescence, and excluding doublets based on a two-dimensional dot plot of forward scatter area (FSC-A) vs. forward scatter height (FSC-H). Doublets have a higher signal width or area-to-height ratio than single cells (singlets), and events deviating from the diagonal are considered doublets. All samples were gently mixed using a vortex mixer to separate agglutinated sperm and then filtered through a 40 μm nylon mesh before acquiring the data. The data were acquired using ImageJ software v1.53n (Rasband, W.S., ImageJ, U.S. National Institutes of Health, Bethesda, MA, USA), and subsequent analysis was performed using FlowJo v10.8.2 (Becton, Dickinson, and Company, Ashland, OR, USA, 2022). Since gender-ablated semen is already stained with Hoechst and red food dye, each flow cytometric analysis was run independently to avoid overlapping emissions. Viability and mitochondrial potential data were adjusted because sperm cells of the undesired sex are inactivated but remain in the straws. The baseline for the adjustment was 2.6 × 10^6^ motile sperm per straw, as explained above. Two technical replicates were conducted with the analysis of a different straw from the same freeze batch each time.

#### 2.4.1. Membrane and Acrosome Integrities

The membrane and acrosome integrities were evaluated following previously described procedures by Anzar et al. and Inanc et al. [31,32,49]. Briefly, diluted sexed semen samples were incubated at 37 °C for 10 min with fluorescent dyes: 1 μL FITC-PNA (Sigma-Aldrich, L7381; stock 1 mg/mL in PBS), 2 μL PI (Molecular Probes, Eugene, OR, USA; P1304MP; stock 2.4 mM in water) per mL semen [32,49] before acquiring the data. The emission and excitation details of all dyes used in this study are shown in Appendix A.

#### 2.4.2. Membrane Stability and Viability

The membrane stability (capacitation rate on the level of sperm membrane) and viability were evaluated following the procedure of Ortega-Ferrusola et al. [30]. Briefly, diluted semen samples (1 mL) were incubated at 37 °C in the dark for 10 min with the addition of 2.6 μL of M540 (Chemodex, St. Gallen, Switzerland; M0033; stock 1 mM in DMSO) and 1 μL of Yo-Pro 1 (Invitrogen, Waltham, MA, USA; Y 3603; stock 25 μM in DMSO) before acquiring the data.

#### 2.4.3. Calcium Level

The intracellular calcium levels were evaluated following the procedure described by Luque et al. [50]. The Fluo-4 AM stock solution was prepared by adding 50 µg Fluo-4 AM (Invitrogen, Waltham, MA, USA; F14201) to 225 µL DMSO for a final concentration of 2 mM. Subsequently, 2.5 μL of the Fluo-4 AM stock solution and 2 μL of PI (2.4 mM water stock) were added per mL of diluted semen and incubated for 20 min at 37 °C. To induce calcium uptake and capture the calcium spike, 5 μL ionomycin (Calbiochem, Ionomycin, calcium salt, Streptomyces conglobatus; 407952; 1 mM stock in DMSO) was added to the samples, and data were acquired one minute after ionomycin addition. The final ionomycin concentration was 4.98 µM.

#### 2.4.4. Mitochondrial Activity

The mitochondrial activity was evaluated according to the procedures of Gravance et al., Ortega-Ferrusola et al., and Spizziri et al. [30,39,51]. Briefly, samples were treated with JC-1 (Invitrogen, Waltham, MA, USA; T3168; stock 2 mM) by adding 7 μL of the stock solution to 1 mL diluted semen. Samples were then incubated for 20 min at 37 °C. Mitochondria stained with JC-1 exhibit fluorescence in both orange and green colors, with the proportion depending on the level of mitochondrial activity. Active mitochondria accumulate JC-1 in their interior and form aggregates that fluoresce orange. In contrast, less functional mitochondria accumulate low concentrations of JC-1, which remain in the monomeric form and fluoresce green. Computer analyses were employed to determine the ratio of JC-1 aggregates to monomers fluorescence for each cell [30,39,51]. Sex-selected semen is already stained with Hoechst, which was used as a gating control to gate out non-Hoechst-stained debris.

### 2.5. Sperm Binding and Interaction with Oviduct Cells

Bovine oviducts were obtained from a local slaughterhouse, transported to the lab in sterile filtered PBS on ice, and washed three times with filtered sterile PBS upon arrival. Oviduct cells were obtained and cultured following the protocol of Medeiros [41]. Briefly, the surrounding connective tissues were trimmed, and epithelial cells were recovered by scraping oviducts with a glass microscope slide into a dish containing PBS (1×) supplemented with 1 µL/mL Gentamicin (stock 50 mg/mL). The epithelial cells were mechanically disrupted by passing them through a 21-gauge syringe needle ten times to eliminate cell aggregates. Afterward, the cells were washed with PBS two times, in addition to a final wash using culture media. Oviduct cells were washed by allowing the cells to settle and then decanting the supernatant without centrifugation. Oviduct cells (five million per dish) were cultured in 35 mm Petri dishes containing 2 mL of equilibrated TCM199 supplemented with 10% (*v*/*v*) fetal calf serum, 50 µg/mL gentamycin, and pyruvate (0.2 mM). The dishes were incubated at 38.5 °C under a 5% CO_2_ in air atmosphere until confluence. Confluence was reached after approximately 6 days of culture. On the day of evaluation, the medium was replaced with equilibrated BGM1 + 6 mg/mL BSA [24] for sperm incubation.

Frozen semen samples were thawed at 37 °C for 30 s, washed two times by centrifugation at 3500× *g* in BGM1, and resuspended to a concentration of 25 × 10^6^ sperm/mL. Forty microliters of the semen suspension containing 1 × 10^6^ sperm cells were added to each dish and incubated for 15 min. The dye YOYO-1 (0.4 µL) was then added with swirling for the identification of dead sperm, with an additional 15 min of incubation. Excitation was set at 380 nm for Hoechst and 495 nm for YOYO, with emission greater than 510 nm for both dyes. Dishes were examined at four time points (30, 180, 420, and 1200 min). At each time point, a dish was washed twice with 1 mL PBS to remove unbound sperm (effluent) and then manually examined using a Nikon Epifluorescent Phase Contrast microscope (Nikon, Minato ku, Japan). Ten fields covering a surface area of 1.65 × 10^6^ µm^2^ were evaluated per dish. Each field contains an average of 1165 oviduct cells. The evaluations included counting all Hoechst-positive sperm in 10 random fields, along with counting dead sperm cells stained with YOYO-1. Oviduct cells do not stain with YOYO-1 due to their viability, and they also do not stain with residual Hoechst. The number of live and bound sperm was then calculated by multiplying the total number of sperm cells in ten random fields by the percentage viability. To obtain non-attached sperm, effluents were centrifuged to concentrate sperm cells, and the number of sperm cells retained in the dish (attached) was determined based on the differences in concentration (evaluated by hemocytometer) between the effluent and the initial. Three technical replicates were conducted with the analysis of a different straw from the same freeze batch each time.

### 2.6. IVF Experiment

Bovine ovaries were sourced from Applied Reproductive Technologies (Madison, WI, USA). The ovaries were transported to the lab, rinsed three times in prewarmed PBS with 10 mL/liter of penicillin–streptomycin (100× Pen/Strep stock with 10,000 IU penicillin and 10 mg streptomycin per mL) to remove excess blood. Cumulus–oocyte complexes (COCs) were obtained by slicing follicles <10 mm in diameter. Oocytes were collected by swirling 12–15 sliced ovaries in 100 mL oocyte collection media, prepared using M199 powder (Gibco, Grand Island, NY, USA) supplemented with bicarbonate (4 mM), Hepes (10 mM), and Pen/Strep (20 mL/liter, stock 100× Pen/Strep). Heparin (2.4 μg/mL) to prevent coagulation and fetal bovine serum (20 mL/liter) were added on the day of use. Subsequently, the resulting mixture was passed through a 100 μm cell strainer and backwashed into a grid plate containing oocyte collection media. Oocytes exhibiting homogeneous cytoplasm and surrounded by at least three layers of cumulus cells were selected under a stereomicroscope. After that, the selected oocytes were washed three times using a sterile 20 μL unopette tip attached to a Hamilton syringe.

Oocytes (50 per well) were matured in 4-well plates containing 500 μL of equilibrated maturation media [52] covered with 300 μL of light mineral oil. Maturation occurred over 24 h at 38.5 °C in a humidified atmosphere of 5% CO_2_ using TCM199 media (with Earle’s salts), supplemented with 10% fetal calf serum, 50 µg/mL gentamicin, 0.2 mM pyruvate, 2 mM L-glutamine, 5 μg/mL follicle-stimulating hormone, and 1 μg/mL Estradiol [53]. Following maturation, 120 oocytes were transferred to 200 μL low bicarbonate TALP in a 1.5 mL microcentrifuge tube to remove cumulus cells through vortex mixing for one minute. Oocytes were then washed in HEPES TALP, placed in 4-well Petri dishes with 60 oocytes added per well, containing 425 μL of IVF-TALP and supplemented with heparin (final concentration of 20 μg/mL) and 20 μL of PHE (stock, 0.5 mM penicillamine, 0.25 mM hypotaurine, and 25 μM epinephrine).

Frozen sperm, thawed for 30 s in 37 °C water, were separated by centrifugation with PureSperm (Irvine Scientific, Santa Ana, CA, USA, 99264) at 3500× *g* for 5 min in 1.5 mL microcentrifuge tubes. Motile sperm cells only penetrate the PureSperm gradient layers (50%/90%) during centrifugation and accumulate at the bottom. Thus, the bottom 100 μL layer was aspirated and washed in 0.5 mL HEPES TALP media by centrifugation at 900× *g* for 3 min. The supernatant was discarded, and the sperm concentration was determined using a hemacytometer. The concentration was adjusted to 25 × 10^6^ per mL in IVF TALP media, and 20 μL of the sperm suspension was added to each fertilization well. The final concentration of sperm cells was 1 × 10^6^/mL, following the procedure by Parrish et al. [44].

For fertilization, cumulus-matured oocytes and sperm were co-incubated at 38.5 °C in a humidified atmosphere of 5% CO_2_ for three different time ranges: 4 h, 8 h, and 12 h. After incubation, free cumulus oocytes were removed, washed three times in HEPES-TALP, and mounted on slides. Cumulus cells were removed prior to fertilization following the procedure by Parrish et al. [44] Oocytes were then carefully pressed under a coverslip supported with a paraffin wax–Vaseline mixture at its corners [54,55], and two sides of the coverslip were sealed with rubber cement. Subsequently, oocytes were fixed in acetic acid–alcohol (1:3) until staining. At the time of examination, oocytes were stained with 1% (*w*/*v*) orcein in 45% (*v*/*v*) acetic acid. Excess stain was cleared with glycerol/acetic acid/water (1:1:3) as described by [56]. The oocytes were then observed under a phase-contrast microscope (400×) for sperm penetration, indicated by a decondensed sperm head with the tail, meiotic stage (anaphase II or beyond), or pronuclei formation [44]. Degenerated oocytes and oocytes with inconclusive observation were excluded. This approach allows for the tracking of fertilizing ability and fertilization progress at each designated time point. The total number of oocytes examined was 649, derived from three technical replicates, with the analysis of a different straw from the same freeze batch each time.

### 2.7. Statistical Analyses

The statistical analyses utilized analysis of variance using mixed models. Time and CLC level were analyzed as the main effects, while the bull effect was treated as a random effect, and time was treated as a repeated measure if two or more time points were included. Percentage data were transformed using arcsine before analysis. Significance was determined at the 5% level for all analyses. Statistical analysis was performed using SAS software (SAS Institute Inc., Cary, NC, USA, SAS 9.4). All values are presented as means or least square means (LSM) with the corresponding standard errors of the mean (SEM).

## 3. Results

### 3.1. Motion Analysis

The results from the CASA evaluations of sperm (n = 4) are presented in Table 1 and Table 2. Treating gender-ablated Jersey semen with CLC at 2 mg/mL of diluted semen containing 67 × 10^6^ sperm cells prior to gender ablation did not increase sperm motility. Only visual motility after 30 min of incubation at 37 °C was significantly higher in the CLC-treated group compared to the control group (54 ± 4 vs. 43 ± 4; *p* < 0.05); see Table 1. The interaction between incubation time and CLC level was not significant for all motility traits (Table 2). The impact of incubation time on sperm motility showed a significant difference (*p* < 0.0001), with total and visual motility exhibiting a difference after 120 min of incubation, while 30 min of incubation was enough to significantly decrease progressive motility.

### 3.2. Flow Cytometric Analyses

Fluorescent data for membrane and acrosome intactness were recorded using photomultiplier detectors and analyzed on a two-dimensional dot plot displaying PI and FITC-PNA intensities. This allowed for the identification of four sperm populations: sperm with intact plasma membrane and intact acrosomes (PI-/FITC-PNA-), sperm with intact plasma membrane and compromised acrosomes (PI-/FITC-PNA+), sperm with compromised plasma membrane and intact acrosomes (PI+/FITC-PNA-), and sperm with compromised plasma membrane and compromised acrosomes (PI+/FITC-PNA+), as described by [31,32,49]. Gating utilized three controls: Hoechst only control (-FITC-PNA, -PI), Hoechst + PI without FITC-PNA, and Hoechst + FITC-PNA without PI. Treating frozen gender-ablated Jersey semen with CLC significantly (*p* < 0.05) increased the adjusted percentage of cells displaying intact membrane and intact acrosome; the control group LSM ± SEM was 28.9 ± 1.2% vs. 34.1 ± 1.2% for the treated group (Table 3).

For data analysis of membrane stability and viability, a Yo-Pro 1/M540 dot plot was utilized to differentiate viable and stable (un-capacitated with low fluidity) plasma membrane (Yo-Pro 1-/M540-), viable and unstable (capacitated with higher fluidity) plasma membrane (Yo-Pro 1-/M540+), and dead (Yo-Pro 1+) events [30]. Gating was performed using three controls: Hoechst only (-M540, -YoPro-1), Hoechst and YoPro-1 without M540, and Hoechst plus M540 without YoPro-1. The adjusted percentage of un-capacitated live cells showed no effect of CLC (*p* > 0.05) with LSM ± SEM of 24.3 ± 6.3% for the control vs. 33.7 ± 6.3% for the CLC-treated group (Table 3). The mitochondrial potential study also showed no difference between the control and CLC-treated groups, with an adjusted orange-to-green ratio of 1.08 ± 0.07 for the control vs. 1.19 ± 0.07 for the CLC-treated group (Table 3).

For analyzing calcium data, the Hoechst-positive population was first selected based on the Hoechst-only stained control (-Fluo-4 AM/-PI). Two additional controls with only PI staining (-Fluo-4 AM/+PI) or only Fluo-4 AM staining (+Fluo-4 AM/-PI) were utilized for gating. Subsequently, for each sample, two-dimensional fluorescence dot plots were generated, and the live (negative for PI staining) and dead (positive for PI) sperm populations were gated. These two populations were employed for the analysis of relative [Ca^2+^]i level, based on the fluorescence intensity of Fluo-4 AM [50]. Relative calcium levels within the live sperm population demonstrated no difference between the CLC-treated and control groups, with an overall mean value of 999 ± 61 for the control vs. 822 ± 61 for the CLC-treated group (Table 4). The CLC-treated sperm cells showed a normal response to acrosome induction (calcium uptake induction), similar to the control group, with both groups exhibiting significantly (*p* < 0.001) higher intracellular calcium levels when induced with ionomycin.

### 3.3. Oviduct Experiment

The study evaluated the binding ability of gender-ablated Jersey frozen sperm cells to oviduct cells in both the control and CLC-treated group at 2 mg CLC/mL diluted semen containing 67 × 10^6^ sperm cells. The total number of live sperm cells bound to a surface area of 1.65 × 10^6^ µm^2^ oviduct cells (10 fields) exhibited no difference between the two groups (*p* > 0.05; Table 5). This indicates a normal binding ability of CLC-treated sperm cells. Furthermore, the interaction between time and CLC level revealed no difference between the groups. The total number of live sperm cells bound to a surface area of 1.65 × 10^6^ µm^2^ gradually decreased over incubation time, with all time points exhibiting differences (*p* < 0.01) from the 30 min time point.

### 3.4. IVF Experiment

The influence of CLC treatment on fertilization rate and pronuclei formation by frozen-thawed gender-ablated Jersey semen was investigated. While there was no difference in fertilization rate between the CLC-treated and control groups after 4 h (58 ± 3% for the control vs. 64 ± 3% for the CLC-treated group) of incubating oocytes with sperm cells, the difference became significant (*p* < 0.05) after 8 and 12 h (Table 6) of co-incubation. At 12 h, which was the endpoint of the experiment, there were 74 ± 3% fertilized oocytes in the control group compared to 82 ± 3% in the CLC-treated group (*p* < 0.05). Both pronuclei formation rates (per total oocytes or fertilized oocytes) showed no difference between the two groups. The polynomial contrast analysis of pronuclei formation rate per fertilized oocytes revealed a linear (*p* < 0.0001) and a quadratic effect (*p* = 0.01) of time. This means that the rate of increase in the percentage of pronuclei formation as a percentage of fertilized oocytes was decreasing, though still increasing, suggesting that it was beginning to plateau. There was a linear increase (*p* < 0.0001) in pronuclei formation as a percentage of total oocytes but no quadratic effect. Thus, pronuclei formation/total oocytes had not yet peaked during the experiment. This indicates that if the experiment had continued for 16 h or more, we might have seen a difference.

The interaction between time and CLC treatment was not significant for all the studied traits. Fertilization and pronuclei formation gradually increased over incubation time (*p* < 0.0001, Table 7). At 12 h post-incubation, about 8.3% (19/230) of the total oocytes in both groups exhibited polyspermy, while 2.6% (6/230) exhibited only one pronucleus, and 5.2% (12/230) exhibited three or more pronuclei. Oocytes with one pronucleus were deemed neither penetrated nor formed pronuclei, as it cannot be guaranteed that this pronucleus originated from a sperm cell. Oocytes exhibiting polyspermy were considered penetrated/fertilized. Oocytes exhibiting three or more pronuclei were classified as polyspermy and included in the pronuclei formation population.

## 4. Discussion

The addition of CLC did not increase sperm motility of post-thaw gender-ablated semen. The motility results agree with those obtained from flow cytometry-based ram sexed semen [57] and bull conventional semen frozen using a programmable freezer [48]. On the other hand, they contradict findings from frozen conventional semen in bull and ram sperm [21,48,58]. Moreover, the CLC treatment did not improve mitochondrial potential, which is required to enhance sperm motility [59]. Cholesterol may have negative effects on mitochondria, which were found to be more pronounced with increasing incubation time [60,61]. This is likely in the case of gender-ablated semen, which undergoes longer processing times compared to conventional semen. Moreover, processing of gender-ablated semen involves highly diluting the samples immediately after CLC treatment. The authors speculated that the higher dilution rate might impact CLC incorporation into the sperm plasma membrane, leading to an attenuated impact. During sexing, sperm cells are exposed to various solutions, pressure, osmolytes, buffers, lasers, and electric charge. How this interacts with CLC is still unknown. Centrifugation, for example, was found to lower the benefits of CLC addition [58,62].

The integrity of the sperm plasma membrane and acrosome is key indicator of successful cryopreservation [47,63]. Compared to the control group, the 2 mg CLC group exhibited a significantly higher percentage of cells with both intact membrane and acrosome, indicating better cryo-survivability with reduced damage to the acrosome. The results align with those of [64,65,66], but contradict the results obtained by [22]. The latter study assessed acrosome reaction using FITC-PNA after inducing calcium uptake with various stimuli, while the present study did not employ any inducers in this experiment. In the calcium experiment of this study, however, ionomycin was used to enhance calcium uptake and was expected to stimulate the acrosome reaction. Both CLC-treated sperm cells and the control exhibited a similar increase in calcium levels, indicating a similar acrosome induction response in both groups, consistent with the findings by [22,39]. Even adding 3 mg CLC/120 × 10^6^ sperm after thawing to sex-selected bovine sperm improved sperm motility and viability over incubation time [67].

The optimum CLC treatment aims to elevate sperm cholesterol levels by 2–3 times (exceeding a 3-fold increase may lead to adverse effects) in bull sperm before cryopreservation [21]. This treatment targets raising the cholesterol/phospholipid ratio (Chol/PL) from 0.45 to >0.8 [20]. Cholesterol/phospholipid ratio (Chol/PL) is the major determinant of sperm plasma membrane fluidity in animals, with bulls exhibiting lower Chol/PL [20]. Capacitated sperm cells lose coating glycoproteins, which increases membrane fluidity and destabilization, allowing more M540 intercalation into the sperm membrane [35,36,37]. Modification of the sperm plasma membrane with cholesterol affects its fluidity and permeability in a temperature-dependent manner. At temperatures above the phase transition (liquid-ordered phase), cholesterol makes the membrane less fluid and thereby less permeable, condensing, stabilizing, and tightening the membrane. Conversely, at subzero temperatures, cholesterol makes the membrane less compact, helping it resist the transition to the solid phase and preventing the lateral separation of phospholipids [20]. Following thawing, sperm cholesterol drastically drops, elucidating capacitation-like changes. However, CLC-treated sperm also lose cholesterol during cryopreservation yet retain a higher cholesterol content than untreated sperm afterward [21,63]. In this study, the M540 intercalation rate showed no response to CLC treatment (*p* > 0.05). Additionally, the downstream capacitation indicators, like intracellular calcium, were not impacted by CLC treatment. Previous reports found that CLC-treated sperm cells had similar calcium levels to untreated cells [22,39]. Additionally, gender ablation factors like high dilution rate and centrifugation may compound the problem as discussed in the previous paragraph.

The oviduct experiment showed no difference between CLC-treated and untreated frozen gender-ablated semen. This contradicts findings from porcine sperm cells treated with CLC, which exhibited a higher interaction rate with oviduct cells [68]. For sperm to interact with oviduct cells, they should have an intact acrosome with proper surface receptors and proteins [43]. Sex-selected sperm are exposed to many physical and chemical stressors, resulting in decapacitation factors removal, loss of acrosome, and lower binding rate to oviductal cells with rapid detachment, all of which indicate higher capacitation-like changes [6,12,14,69]. Compared to non-sex-selected semen, sex selection was found to alter the composition of the sperm plasma membrane, reduce surface proteins, mitochondrial potential, and acrosome integrity, and induce changes in glycolytic and oxidative phosphorylation enzymes. It may also affect sperm–egg fusion proteins, reduce sperm motility and the proportion of hyperactivated sperm, impair fertilization, and even lead to failure in supporting acceptable early embryonic development [12,13,70]. It seems that CLC treatment cannot alleviate these alterations to receptors and surface proteins, resulting in similar binding ability to oviduct cells.

The CLC addition in the present experiment improved membrane integrity and acrosome status, which are reflected in the better penetration rate in the IVF experiment. Even adding CLC to sex-selected semen after thawing was reported to improve the fertilizing ability of sperm and embryonic development up to the late blastocyst stage [71]. Capacitation is judged to be complete when sperm can penetrate a zona-intact oocyte, fuse with the oolemma of the oocyte, and decondense in the oocyte cytoplasm [24,44]. The rate of sperm penetration, decondensation, and pronuclei formation was assessed in the present study to reflect the capacitation window and help in determining the time of fertilization [44,45], giving us insights into how CLC-treated cells respond to capacitation induction in vitro. The CLC-treated group showed no delays in capacitation compared with the control group. A study was conducted at Colorado State University to evaluate the capacitation time of conventional CLC-treated sperm using a flow cytometry approach. Their findings indicated that the addition of cholesterol resulted in a one-hour delay in capacitation compared to the untreated control [65]. In the IVF work in our study, oocytes were checked every 4 h, limiting our ability to detect treatment differences in capacitation time less than 4 h.

Future studies might focus on the interaction between CLC and sexing and cryopreservation chemicals and conditions. Especially since cholesterol alters the fluidity of the sperm plasma membrane, adjustments to the extender composition may be necessary, considering membrane permeability after cholesterol addition. For example, reducing glycerol from 6% to 3% when CLC is utilized improved post-thaw sperm quality and reduced the toxic impacts of glycerol on DNA [72]. This adds one more benefit to sex-selected semen when CLC is used, as sex-selected semen exhibits a higher DNA damage rate when compared to conventional semen [13,70]. Furthermore, the addition of CLC can effectively replace egg yolk, giving comparable cleavage and blastocyst rates [49]. Egg yolk is essential for sperm viability during sexing, but it cannot be added at higher levels before sex selection. The final egg yolk level is only 2% before sorting, probably to avoid clumps in the sexing machine. Cholesterol could provide an alternative option, which resulted in higher post-thaw sperm viability in the present study [44]. The impact of cholesterol supplementation on the dynamics of membrane domains, lipid rafts, surface receptors, and enzyme activity within sperm before cooling and cryopreservation or upon thawing still needs further investigation. Lastly, this study did not directly measure the sperm cholesterol content after treatment and thawing due to funding constraints. However, in future studies, we plan to validate flow cytometry-based methods using Filipin III [62] or track cholesterol efflux with BODIPY-cholesterol [28] in bovine sperm.

Sperm capacitation times may vary by only one-hour increments [65], while the IVF work was checked every 4 h, which limited our ability to detect treatment differences in capacitation times less than 4 h. Thus, we can detect the delay, but it is challenging to determine the exact time of change. However, since we can distinguish between the rates of various stages of fertilization over the designated time points (penetrated sperm, decondensed sperm head, and sperm pronuclei formation), we should be able to determine whether the delay occurred or not. One limitation of the IVF experiment is that the time required to capacitate sperm in vitro is not necessarily that needed to capacitate sperm in vivo. However, we expect the capacitation times to be relative, and if CLC-treated sperm require more time to capacitate in vitro, we would also expect these sperm to require longer capacitation times in vivo. Further investigation, including a fertility trial, is still needed, taking into consideration the possible delay in the in vivo capacitation window of treated sperm cells [20,63,73]. Another limitation is that we cultured these embryos for only 12 h, which aligned with our objective at this stage. In future studies, we should consider culturing them to the blastocyst stage or even transferring them to recipient cows.

## 5. Conclusions

The utilization of CLC added a minimal improvement to the post-thaw quality of gender-ablated semen. It did not significantly improve either the sperm motility or the capacitation status. However, the sperm population that has intact membranes and acrosomes was significantly increased. Mitochondrial potential and ability to interact with oviduct cells were maintained. The CLC treatment did not delay capacitation progress. However, the fertilization rate was significantly improved over time. Further studies are still required to maximize the benefits of CLC treatment, which may focus on the interaction of CLC with egg yolk, glycerol, and other extender components and sexing factors. Future studies may also explore adding other molecules like fatty acids, not just cholesterol, to achieve a balanced treatment with better cryopreservation outcomes. As well as a fertility trial to fully validate the overall efficacy.

## Figures and Tables

**Table 1 animals-15-02038-t001:** Adjusted ^a^ sperm motility of Jersey gender-ablated frozen-thawed semen treated with cholesterol-loaded cyclodextrin (CLC) ^b,c^.

Post-Thaw Time (min)	CLC-Level (mg/67 × 10^6^ Sperm Cells)	Visual Motility (%)	Total Motility via CASA (%)	Progressive Motility via CASA (%)
0	0	44 ± 4	39 ± 4	19 ± 3
	2	54 ± 4	49 ± 4	23 ± 2
30	0	43 ± 4	37 ± 3	10 ± 1
	2	54 * ± 4	45 ± 3	14 ± 1
60	0	36 ± 7	32 ± 6	8 ± 2
	2	50 ± 7	45 ± 6	11 ± 2
120	0	18 ± 4	18 ± 3	3 ± 2
	2	23 ± 4	21 ± 3	4 ± 1

^a^ The motility data were adjusted as the Y-bearing sperm cells were inactivated without separation, so motility was calculated as a proportion of the initial 2.6 × 10^6^ motile X-bearing sperm cells per packed straw. ^b^ The CLC was added before gender ablation at level 2 mg/mL semen containing 67 × 10^6^ sperm cells compared to no CLC (n = 4, 2 technical replicates). ^c^ Motility was assessed at four time points (0, 30, 60, and 120 min) post-thaw. All values are expressed as least square means ± standard error mean. The main effect of CLC was not significant for any motility trait. Asterisks within a column at each time point indicate statistically significant differences between CLC levels for that time (*p* < 0.05).

**Table 2 animals-15-02038-t002:** Repeated measures analysis of time as a factor affecting adjusted sperm motility ^a^.

Post-Thaw Time (min)	Visual Motility (%)	Total Motility via CASA (%)	Progressive Motility via CASA (%)
0	50 ± 4	45 ± 4	21 ± 2
30	50 ± 4	42 ± 3	12 *** ± 1
60	45 ± 7	40 ± 6	10 *** ± 2
120	21 *** ± 5	20 *** ± 4	4 *** ± 1

^a^ The motility data were adjusted as the Y-bearing sperm cells were inactivated without separation, so motility was calculated as a proportion of the initial 2.6 × 10^6^ motile X-bearing sperm cells per packed straw. Data include both the CLC-treated and control groups. Asterisks within a column indicate significant differences compared to the 0 min time point. Values are expressed as means ± SEM. Time exhibited a significant effect (*p* < 0.0001) for all motility traits, while the interaction between CLC and time was not significant for any trait.

**Table 3 animals-15-02038-t003:** Flow cytometric analyses of gender-ablated frozen-thawed Jersey semen treated with cholesterol-loaded cyclodextrin (CLC) ^a^.

CLC Level (mg/67 × 10^6^ Sperm Cells)	Adjusted ^b^ Sperm Cells with Intact Membranes and Acrosomes (%)	Adjusted ^b^ Live Un-Capacitated Sperm Cells (%)	Mitochondrial Potential (Adjusted ^b^ Ratio of JC1 Orange to Green Florescence)
0	28.9 ± 1.2	24.3 ± 6.3	1.08 ± 0.07
2	34.1 * ± 1.2	33.7 ± 6.3	1.19 ± 0.07

^a^ CLC was added prior to gender ablation at level 2 mg/mL diluted semen containing 67 × 10^6^ sperm cells compared to no CLC (n = 4, two technical replicates). Data is presented as the least square mean ± standard error mean. ^b^ Data were adjusted as the Y-bearing sperm cells were inactivated without separation, so motility was calculated as a proportion of the initial 2.6 × 10^6^ motile X-bearing sperm cells per packed straw. Asterisks (*) within a column indicate statistically significant differences between the control and CLC-treated groups (*p* < 0.05).

**Table 4 animals-15-02038-t004:** Relative intracellular calcium flow cytometric analyses of gender-ablated frozen-thawed Jersey semen treated with cholesterol-loaded cyclodextrin (CLC) *.

CLC Level (mg/67 × 10^6^ Sperm Cells)	No Ionomycin	Ionomycin	Overall
0	627 ± 102	1370 *** ± 102	999 ± 61
2	482 ± 102	1162 *** ± 102	822 ± 61
Overall	554 ± 173	1266 ± 173	

* CLC was added prior to gender ablation at level 2 mg/mL diluted semen containing 67 × 10^6^ sperm cells compared to no CLC (n = 4, two technical replicates). Asterisks, ***, within a column indicate statistically significant differences between the CLC-treated and control groups, *p* < 0.0001.

**Table 5 animals-15-02038-t005:** Impact of cholesterol-loaded cyclodextrin (CLC) on the binding ability of gender-ablated frozen Jersey sperm cells to oviduct cells ^a^.

Post-Incubation Time (min)	CLC-level (mg/67 × 10^6^ Sperm Cells)	Number of Viable Bound Sperm Cells/Area of 1.65 × 10^6^ µm^2^ Oviduct Cells	Overall
30	0	533 ± 48	537 ± 99
	2	541 ± 48	
180	0	336 ± 52	352 ** ± 36
	2	367 ± 52	
420	0	219 ± 28	221 ** ± 19
	2	222 ± 28	
1200	0	66 ± 18	74 ** ± 12
	2	82 ± 18	

^a^ CLC was added prior to gender ablation at 2 mg/mL diluted semen containing 67 × 10^6^ sperm cells compared to no CLC (n = 4, three technical replicates). Data was collected at various time points (30, 180, 420, and 1200 h) post-incubation with sperm cells in the dish. Overall means across both the CLC-treated and control groups were calculated for each time point. Asterisks (**) indicate statistically significant differences compared to the 30 min time point, *p* < 0.01.

**Table 6 animals-15-02038-t006:** In vitro fertilizing ability and pronuclei formation rate of gender-ablated frozen Jersey sperm cells treated with cholesterol-loaded cyclodextrin (CLC) *.

Post-Incubation Time (h)	CLC-Level (mg/67 × 10^6^ Sperm Cells)	Fertilization (%)	Pronuclei Formation/Fertilized Oocytes (%)	Pronuclei Formation/Total Number of Oocytes (%)
4	0	58 ± 3	1 ± 1	1 ± 1
	2	64 ± 3	0 ± 1	0 ± 1
8	0	67 ± 3	37 ± 3	25 ± 3
	2	77 * ± 3	38 ± 3	29 ± 3
12	0	74 ± 3	59 ± 4	43 ± 3
	2	82 * ± 3	61 ± 4	50 ± 3

* CLC was added before gender ablation at 2 mg/mL diluted semen containing 67 × 10^6^ sperm cells compared to the control with no CLC (n = 4, with three technical replicates). The total number of examined oocytes was 649. Data is presented as least square means ± standard error mean. Asterisks within a column at each time point indicate statistically significant differences, *p* < 0.05.

**Table 7 animals-15-02038-t007:** Repeated measures analysis of time as a factor affecting fertilizing ability and pronuclei formation rate of gender-ablated semen *.

Post-Incubation Time (h)	Fertilization (%)	Pronuclei Formation/Fertilized Oocytes (%)	Pronuclei Formation/Total Number of Oocytes (%)
4	61 ± 2	1 ± 1	0 ± 0
8	72 *** ± 2	38 *** ± 2	27 *** ± 2
12	78 *** ±2	60 *** ±3	46 *** ± 2

* Data include both the CLC-treated and control groups. Asterisks within a column indicate significant differences compared to the four-hour time point (***, *p* < 0.001). Values are expressed as means ± SEM. Fertilizing ability and pronuclei formation rate increased linearly (*p* < 0.0001) over time. The interaction between CLC level and time was not significant for any trait, and thus, data were pooled for CLC-treated and control groups.

## Data Availability

The original contributions presented in this study are included in the article. Further inquiries can be directed to the corresponding author.

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
