# Peer review of "Using Cholesterol-Loaded Cyclodextrin to Improve Cryo-Survivability and Reduce Capacitation-Like Changes in Gender-Ablated Jersey Semen"

_animals, 2025, doi:10.3390/ani15142038_

Round 1
Reviewer 1 Report
Comments and Suggestions for Authors
The manuscript:” Using Cholesterol-Loaded Cyclodextrin to Improve Cryo-survivability and Reduce Capacitation-Like Changes of Gender-Ablated Jersey Semen” is well written and report in a clear way the goals and the experiments ran to verify the hypothesis behind the aims. The topic in se is very important for all the livestock industry, It is a pity the small number of bulls evaluated, a higher number could have given more emphasis to your founds. Furthermore, I could have found some differences if compared with conventional semen.
Few minors:
Line 84 please change the 75% with 25 % it will make the sentence clearer
Line 190 please add the conc of Hoechst used
Line 204 maybe could be better moving that sentence before 201 line.
Line 322 could be possible to report the breeds of the ovaries used.
Line 325 never read before massaged the ovaries…..
Line 322 please specify the grade of the oocytes selected: following international standards
Line 353 could please explain why you fertilized with a final conc of 1.1 million sperm per ML: 425 ul IVF + 20 of PHE + 20 of sperm (conc 25*10^6). Furthermore, change mL in ul (line 434)
Line 627 it should be mentioned as greatest limitation of the IVF not the timing but rather the embryo production up to blastocyst stage or more to the embryo transfer……
Please remove the p value all the time that there is NOT statistical significance
Major:
Please could you put in each table the row number in parenthesis and not only the percentage
From line 341 it is really hard what did you do. You took 120 COC and removed the cumulus and then what happened to these? Line 355 you reported a co-incubation of COC and sperm, did you mount on slide the COC, without removing the cumulus?
It will be really important to rewrite this section and the total number of oocytes in the manuscript not only in the legend of table 5A
Author Response
Reviewer 1
Comment 1: (The topic in se is very important for all the livestock industry, It is a pity the small number of bulls evaluated, a higher number could have given more emphasis to your founds. Furthermore, I could have found some differences if compared with conventional semen.)
Response: Thank you for pointing this out. This was done with Jersey bulls due to funding requirements, and the number was limited.
Comment 2: (Line 84 please change the 75% with 25 % it will make the sentence clearer.)
Response: We respectfully disagree. Based on the data we provided, the fertility of gender-ablated semen is approximately 75% of that obtained with conventional semen.
Comment 3: (Line 190 please add the conc of Hoechst used.)
Response: This is a trade secret and cannot be disclosed; therefore, we stated that it was performed according to ABS's standard operating procedures (line 245) and proprietary in line 247.
First few words: amount proprietary to ABS
Comment 4: Line 204 maybe could be better moving that sentence before 201 line.
Response: Thank you for pointing this out. The revision has been made, and the changes are reflected in lines 262–272.
First few words: During gender ablation, sperm cells of the undesired sex were inactivated but remained in the sample [8].
Comment: Line 322 could be possible to report the breeds of the ovaries used.
Response: We obtained these ovaries from a company that sources them from a slaughterhouse in Green Bay; therefore, we expect them to be from cows of various breeds.
Comment: Line 325 never read before massaged the ovaries…..
Response: Done; see line 415.
First few words: to remove excess blood
Comment: Line 322 please specify the grade of the oocytes selected: following international standards
Response: Done; see lines 422-424
First few words: oocytes exhibiting homogenous
Comment: Line 353 could please explain why you fertilized with a final conc of 1.1 million sperm per ML: 425 ul IVF + 20 of PHE + 20 of sperm (conc 25*10^6). Furthermore, change mL in ul (line 434)
Response: Done; see lines 440 and 451. The sperm concentration was determined according to the protocol established by Dr. Parrish in our lab.
First few words: microliter in line 440; and the final concentration of sperm cells…..
Comment: Line 627 it should be mentioned as greatest limitation of the IVF not the timing but rather the embryo production up to blastocyst stage or more to the embryo transfer……
Response: Done; see lines 813–816. We would also like to clarify that, in alignment with our goal of determining the timing of capacitation or detecting any delays, our original plan, even during the grant proposal stage, was only to investigate whether cholesterol addition causes delays in capacitation. In future work, we may consider extending embryo development to the blastocyst stage or even performing embryo transfer, as you suggested. For now, our focus is simply to assess whether the sperm behave normally.
First few words: Another limitation…..
Comment: Please remove the p value all the time that there is NOT statistical significance
Response: Thank you for pointing this out. The change has been applied throughout the entire paper.
Comment: Please could you put in each table the row number in parenthesis and not only the percentage. From line 341 it is really hard what did you do. You took 120 COC and removed the cumulus and then what happened to these? Line 355 you reported a co-incubation of COC and sperm, did you mount on slide the COC, without removing the cumulus? It will be really important to rewrite this section and the total number of oocytes in the manuscript not only in the legend of table 5A.
Response: This section has been rewritten and is now clearer; see lines 455-469. I have specified that motility data were collected from at least 500 sperm cells. Flow cytometry was performed on 10,000 Hoechst-positive cells after excluding doublets, and a total of 649 oocytes were examined. Raw numbers are typically not reported for motility or flow cytometry data, I have not seen this done in the literature. For the IVF data, analysis was conducted using SAS, with bulls included as a random factor and time as a repeated measure. This is the appropriate method for analyzing such data.
First few words: free cumulus oocytes were removed…… until the end of the paragraph
Reviewer 2 Report
Comments and Suggestions for Authors
Highlights: not necessary as per journal guidelines
Line 42: add what are the parameters checked in this experiment, abstract should typically contain intro, materials and methods, results and conclusions
line 134: no need to write your results in the introduction
line 194: how conc. of sperm to add CLC was chosen
Line 397: how data is for both treatment and control, it shows only one value here in table at diff times 0, 30. 60, 120
statistical glitches as mention in the manuscript has to be addressed
All minor corrections are given in manuscript

Author Response
Reviewer 2
Comment: Line 42: add what are the parameters checked in this experiment, abstract should typically contain intro, materials and methods, results and conclusions.
Response: Thank you for pointing this out. The adjustment has been made; please see lines 33-35.
First few words: were evaluated and adjusted to…
Comment: line 134: no need to write your results in the introduction
Response: It was removed; see line 183.
First few words: Cryopreservation may lead to
Comment: line 194: how conc. of sperm to add CLC was chosen
Response: Done; see lines 256 -260. This is the company’s (ABS) standard protocol. They first dilute the samples to 200 million for staining, then further dilute them to 67 million prior to sexing. This procedure cannot be adjusted on our end.
First few words: the previous CLC level….
Comment: Line 397: how data is for both treatment and control, it shows only one value here in table at diff times 0, 30. 60, 120. statistical glitches as mention in the manuscript has to be addressed.
Response: Statistical glitches have been addressed. However, we want to clarify that Table 1A presents the data layout for both the treatment and control groups. Table 1B, as indicated in its title, pertains specifically to the repeated measures analysis. Time is the only factor treated as a repeated measure in this analysis, as we collected data at four distinct time points. Moreover, the interaction between time and CLC was not significant. It’s important to note that if time had not been analyzed as a repeated measure, it would have violated the assumptions of ANOVA, particularly the assumption that each observation is independent. When observations are collected from the same experimental units across multiple time points, time must be modeled as a repeated measure.
Bulls were included as a random factor to allow our conclusions to be generalized beyond the specific bulls used in the study. If bulls were treated as a fixed factor, any conclusion would only apply to those particular bulls. Including bulls as a random factor reduces the power to detect significance but provides a more appropriate and generalizable analysis.
Given that bulls were treated as a random factor and time was analyzed as a repeated measure, a mixed model was the correct analytical approach, rather than a general linear model (GLM). We consulted multiple statisticians in both the Department of Dairy and Animal Sciences and the Department of Statistics at UW–Madison, and all confirmed that this is the appropriate method for analyzing this type of data. Lastly, three asterisks (***) in the tables indicate that the p-value was less than 0.001.
Reviewer 3 Report
Comments and Suggestions for Authors
-
Dear the editor/the authors,
After carefully reading the manuscript entitled “Using Cholesterol-Loaded Cyclodextrin to Improve Cryo-survivability and Reduce Capacitation-Like Changes of Gender-Ablated Jersey Semen” I’d like to report my review as follow;
Strength: This manuscript was prepared with tremendous efforts. The authors thoroughly employed a variety of techniques to determine the putative effects of CLC treatment on the gender-ablated spermatozoa from simple CASA analyses to fertilization ability through IVF. To my view, it was also a rare opportunity to conduct an experiment inside the ABS sorting facility headquarter despite for the benefit of the company mainly.
Weakness: This manuscript also contains slight weaknesses in which the improvement was minimal compared to the enhancement of cryosurvival reported with unsorted spermatozoa in several species. One doubt was the sonication of CLC working solution that has never been practiced before. The effect of sonication on the intact of CLC was unknown. Traditionally, sonication is only used during the preparation of CLC. However, as mentioned in the Materials and Methods (line 180), it was necessary as a safety precaution to prevent clogging of the sexing apparatus since this apparatus utilizes the stream of microfluid as opposed to microdroplets utilized by its sexing counterpart.
Another weakness is that the amount of cholesterol in the CLC-treated spermatozoa was not directly determined. On the other hand, the authors used an indirect approach, as mentioned in the Limitations of the study (line 606-610), to convince the uptake of a certain amount of cholesterol by spermatozoa. It would be nice to see how much cholesterol was incorporated into spermatozoa as reported in some publications especially when the modifications were made from the well-accepted protocol.
Specific comments
No comment at all, well written.
Author Response
Reviewer 3
Comment: (Weakness: This manuscript also contains slight weaknesses in which the improvement was minimal compared to the enhancement of cryosurvival reported with unsorted spermatozoa in several species. One doubt was the sonication of CLC working solution that has never been practiced before. The effect of sonication on the intact of CLC was unknown. Traditionally, sonication is only used during the preparation of CLC. However, as mentioned in the Materials and Methods (line 180), it was necessary as a safety precaution to prevent clogging of the sexing apparatus since this apparatus utilizes the stream of microfluid as opposed to microdroplets utilized by its sexing counterpart.)
Response: Thank you so much for pointing this out. As mentioned in the Materials and Methods section, we conducted a preliminary study to evaluate the impact of sonication. The data are shown in the figure below and indicate no significant differences between sonication and non-sonication of the working solution.
This was one of the preparatory experiments conducted prior to implementing CLC into the gender-ablated semen protocol. This study focused on the impact of sonication of the working solution on post-thaw sperm visual motility, total and progressive motilities by CASA (n = 4). This experiment was conducted using conventional semen, with cholesterol-loaded cyclodextrin applied at levels of 2 mg and 4 mg per mL of semen containing 120×10⁶ sperm cells. Motility was assessed at three time points (0, 0.5, and 1 hour) and then averaged. All values are expressed as means ± standard error of the mean (SEM).
Comment: (Another weakness is that the amount of cholesterol in the CLC-treated spermatozoa was not directly determined. On the other hand, the authors used an indirect approach, as mentioned in the Limitations of the study (line 606-610), to convince the uptake of a certain amount of cholesterol by spermatozoa. It would be nice to see how much cholesterol was incorporated into spermatozoa as reported in some publications especially when the modifications were made from the well-accepted protocol.)
Response: We agree and will consider this in future studies, as noted in the limitations section.
Round 2
Reviewer 1 Report
Comments and Suggestions for Authors
Good to go!
Reviewer 2 Report
Comments and Suggestions for Authors
Nil